# Butterfly Wing Translucence Enables Enhanced Visual Signaling

**DOI:** 10.3390/insects14030234

**Published:** 2023-02-26

**Authors:** Doekele G. Stavenga, Heinrich L. Leertouwer, Kentaro Arikawa

**Affiliations:** 1Groningen Institute for Evolutionary Life Science, University of Groningen, NL9747AG Groningen, The Netherlands; 2Research Center for Integrative Evolutionary Science, Sokendai-Hayama, The Graduate University for Advanced Studies, SOKENDAI, Hayama 240-0193, Japan

**Keywords:** pigmentary coloration, structural coloration, papilionids, bile pigments, reflection, transmission

## Abstract

**Simple Summary:**

The commonly held view on the coloration of butterfly wings is that the bright colors of the dorsal wing sides are used for display, while the much duller ventral wing sides provide camouflage, specifically when the butterflies are resting. Here, we show that the wings of many butterfly species are so transparent that the light transmitted by the wings, especially during flight, will contribute to butterflies’ visibility.

**Abstract:**

The light reflected by the dorsal side of butterfly wings often functions as a signal for, e.g., mate choice, thermoregulation, and/or predator deterrence, while the ventral wing reflections are generally used for crypsis and camouflage. Here, we propose that transmitted light can also have an important role in visual signaling because, in many butterfly species, the dorsal and ventral wing sides are similarly patterned and locally more or less translucent. Extreme examples are the Japanese yellow swallowtail (*Papilio xuthus* Linnaeus, 1758) and the Yellow glassy tiger (*Parantica aspasia* Fabricius, 1787). Their wings exhibit a similar color pattern in reflected and transmitted light, which allows enhanced visual signaling, especially in flight. Contrasting cases in which the coloration and patterning of dorsal and ventral wings strongly differ are the papilionid *Papilio nireus* Linnaeus, 1758, and the pierid *Delias nigrina* Fabricius, 1775. The wings observed in reflected or transmitted light then show very different color patterns. Wing translucence thus will strongly affect a butterfly’s visual signal.

## 1. Introduction

Butterflies are universally admired for their colorful patterns displayed by the dorsal side of their wings. In many butterfly species, color patterns are used for sexual signaling, especially when males are more brightly colored than females [1,2,3,4]. In distasteful and noxious species, the bright dorsal wing sides are also used for aposematic signaling [5]. In contrast, ventral wing sides are often inconspicuously colored, which is assumed to be used for predator avoidance when the butterfly is resting with closed wings, thus supporting camouflage or crypsis [6].

Butterfly wings consist of a wing membrane that is commonly covered on both sides with scales, like shingles on a roof. The scales are divided into cover and ground scales that partially overlap. The cumulative optical processes of reflection, absorption, and transmission of various scales, as well as the wing membrane proper, determine the final wing color [7,8,9,10]. Butterfly wing scales are commonly colored by pigments: pterins in pierids [11], ommochromes in nymphalids [12,13], and papiliochromes in papilionids [14,15]. As the scales are thin, pigment absorbance remains low (generally well below 1), except in some scales that contain a high concentration of the ubiquitous brown-black melanin. Furthermore, the wing membranes of several butterfly species are pigmented by carotenoids and bile pigments [16]. Butterfly wing scales can also have a structural color, which can be caused by a variety of structures, e.g., the scale lower lamina, acting as an optical thin film, multilayers in the scale lumen or the scale ridges, or complex photonic structures such as gyroids [10,17,18,19,20,21]. A backing of melanin is commonly in place to augment the structural color, as the reflected light would be easily washed out due to light scattering from other scales and nonselective wavelength reflections of the wing membrane [22].

In our studies on the pigmentary and structural coloration of butterfly wings, we noticed that the dorsal and ventral wing sides of several butterfly species were, on the one hand, quite reflective but, on the other hand, also somewhat translucent. Moreover, in many butterfly species, the dorsal and ventral wing sides appeared to be similarly colored and patterned, which suggested that in those cases, the ventral wing coloration is also part of the butterfly’s display. For example, in the common bluebottle *Graphium sarpedon* Linnaeus, 1758 [23] and the great emperor butterfly *Sasakia charonda* Hewitson, 1863 [24], the dorsal and ventral wing patterns strikingly resemble each other, and in transmitted light, virtually the same patterning is observed as that in reflected light. The possible contribution of transmitted light to wing coloration has so far received little attention, although it may play an important role in a butterfly’s visual signaling. Here, we investigate various butterfly species to unravel the factors that control wing translucence, and we discuss the consequences of the similar patterning of a butterfly’s dorsal and ventral wings for the animal’s visibility.

## 2. Materials and Methods

### 2.1. Specimens and Photography

Mounted specimens, purchased from various commercial sources (Worldwide Butterflies Ltd., (Dorset, UK), thebugmaniac.com, demuseumwinkel.com), were photographed with a Nikon D70 digital camera, equipped with an F Micro-Nikkor lens (60 mm, f2.8; Nikon, Tokyo, Japan). The reflected-light images were obtained by applying epi-illumination of dorsal and ventral wing sides. The transmitted-light images were obtained by illuminating the butterflies’ ventral wings and photographing the dorsal wing sides. Only RGB (true-color) photos are presented, as UV reflectance and transmittance are generally low. Photographs of butterflies in their natural habitat were taken with a Canon 6D Mark II digital camera, equipped with an EF Macro lens (100 mm, f2.8; Canon, Tokyo, Japan). (Reflectance and transmittance are physical quantities, i.e., the fraction of incident light that is reflected and transmitted, respectively; absorbance is the −log10 of the transmittance.)

### 2.2. Spectrophotometry

Reflectance spectra of local wing areas (size ~1 mm^2^) were measured with a bifurcated probe and an Avantes AvaSpec-2048-2 CCD detector array spectrometer (Avantes, Apeldoorn, The Netherlands). The reference standard was a white diffuser (Avantes WS-2). Transmittance spectra were measured with an integrating sphere (AvaSphere-50, Avantes) and the detector array spectrometer.

## 3. Results

We surveyed the butterfly species in our collection by photographing their dorsal wing sides in both reflected (Figure 1A–L; DR) and transmitted (Figure 1A–L; DT) light. We only show photographs of the ventral side in reflected (Figure 1A–L; VR) light, because the transmission patterns of both wing sides were identical. Figure 1 presents several papilionid, nymphalid, and pierid butterflies.

An outstanding example in which the reflection images of both sides are very similar and also virtually identical to the pattern observed in transmitted light is the well-studied Japanese yellow swallowtail, *Papilio xuthus* [25]. On both wing sides, melanized veins separate cream-yellow areas in virtually the same pattern [26]. The spectral characteristics of the creamy patches are given in Figure 2A. The reflectance spectrum of the dorsal wing side (DR) shows a higher modulation than that of the ventral side (VR), but all spectra are dominated by the absorption spectrum of the pigment papiliochrome II, which is exclusively absorbed in the ultraviolet-to-violet wavelength region, causing the cream color [15]. The transmittance spectra obtained by illuminating the dorsal or ventral side are identical in measurement accuracy (Figure 2A; DT and VT).

The wings of the congeneric green-banded swallowtail, *Papilio phorcas* (Figure 1B), are dorsally colored by a band of greenish scales, which cover a green-pigmented wing membrane. The ventral side of the dorsal green bands is whitish due to a cover of white scales. In transmitted light, however, the *P. phorcas* wings virtually display an identical color pattern as the pattern that is dorsally observed in reflected light. Figure 2B shows the reflectance and transmittance spectra of the central, colored bands. The reflectance spectrum measured from the dorsal side (DR) shows a pronounced depression, at 605 nm, due to bile pigment [27]. The reflectance spectrum of the ventral side (VR) is much higher and hardly shows any depression, evidently due to the strong, broadband scattering by the ventral white scales. The effect of the white scales can be understood from the fact that dorsally incident light is partly reflected and partly traverses firstly the scales of the dorsal wing side and subsequently the colored wing membrane. The latter fraction is then partly backscattered by the ventral white scales. After traveling back through the pigmented wing membrane and dorsal scales, it leaves the dorsal wing side again and adds to the reflection, thus enhancing the green color. The transmittance spectra of both sides (Figure 2B; DT and VT) are identical and show a broad green band with a slight depression at 603 nm.

The wings of the common bluebottle swallowtail, *Graphium sarpedon*, have a similar arrangement as *P. phorcas* and show a very similar patterning of both wing sides [23]. The dorsal wing side of the purple-spotted swallowtail, *Graphium weiskei* (Figure 1C) displays an unusual colorful pattern, consisting of purple, green, and blue patches, due to a combination of bile pigments and carotenoids [28]. On the ventral side of the purple dorsal area, backscattering white scales are again found, obscuring there the purple-colored wing membrane when observing the ventral side while applying ventral illumination. Yet, keeping this ventral illumination and observing the dorsal side in transmitted light, the same pattern is seen as that in reflected light, i.e., when applying dorsal illumination. Essentially the same correspondence of reflection and transmission patterns is encountered in other papilionids, although the pigmentation may be slightly different (e.g., *G. milon*; Figure 1D).

The correspondence of reflection and transmission images is less striking in Rothschild’s birdwing (*Ornithoptera rothschildi*; Figure 1E), in which only the hindwings have a few translucent patches. The African blue-banded swallowtail (*Papilio nireus*; Figure 1F) shows even stronger differences, as the colored dorsal bands are only vaguely visible in transmitted light, clearly because its wings and/or ground scales are heavily melanized. As we will discuss below, this is most likely related to the structural coloration displayed by the dorsal cover scales.

In nymphalids, the same gradient in patterning and translucence can be recognized (Figure 1G–I). The yellow glassy tiger, *Parantica aspasia* (Figure 1G), a danaid butterfly, shows a saturated-yellow hindwing as well as a partly colored forewing together with a framework of melanized veins. The framed areas are devoid of scales, and the yellow color is due to a carotenoid [16] that is concentrated in the wing membrane, resulting in identical wing patterns seen in reflected and transmitted light. For two other nymphalids, the pearl emperor, *Charaxes varanes* (Figure 1H), and speckled wood, *Pararge aegeria* (Figure 1I), the dorsal and ventral reflection patterns differ somewhat, but the transmitted light patterns of the dorsal wings are again very similar to those seen in reflected light.

In pierids, various patterns are observed (Figure 1J–L). The dorsal and ventral reflection patterns of the crimson tip, *Colotis ione* (Figure 1J), and the clouded yellow, *Colias croceus* (Figure 1K), rather differ, but their transmitted light patterns are again similar to the dorsal reflection patterns, showing that the translucence of the ventral yellow scales is high. The dorsal hindwings of *C. croceus* and the dorsal forewing tips of *C. ione* are iridescent, due to multilayered scales, with reflection bands in the UV and blue-wavelength region, respectively [29,30], but the iridescence is only seen with about specular-reflected light, so generally, the dorsal reflection and transmission images will be similar. An extraordinary case is made by *Delias nigrina* (Figure 1L). The dorsal wings closely resemble those of other pierids, but ventrally, a large area of melanized scales is interrupted by only a few narrow bands of yellow and red scales [8,11]. As in *P. nireus*, the transmitted-light image is dominated by the strongly absorbing melanized scales, while the colored bands only vaguely shine through (Figure 1F,L). In these cases, the transmitted-light pattern rather resembles the reflected-light pattern of the ventral wing side.

Figure 3 shows a number of butterfly species, namely nymphalids, lycaenids, papilionids, and pierids, with very different wing patterns but with considerable wing transmittance, photographed in their natural habitat. The photographs indicate that translucent wings will also appear on and off during flight.

## 4. Discussion

### 4.1. Butterflies with Translucent Wings

Our survey of several papilionids as well as species from other butterfly families show that the dorsal and ventral sides of the wings feature quite similar patterns. In some species, the patterning is virtually identical; thus, due to translucence, the color pattern of the dorsal wings seen in reflected light is very similar to the pattern observed in transmitted light (Figure 1A,G). Figure 4 presents additional examples of papilionids with translucent wings (*Graphium sarpedon, G. stresemanni*, *G. antiphates*, *G. agamemnon*, *Papilio hesperus*, and *P. ophidicephalus*) and cases with nontranslucent wings (*P. bromius*, *P. epiphorbas*).

Many butterfly species from other families also show very similar wing patterns in reflection and transmission, viz. *Nessaea hewitsonii*, *Charaxes subornatus*, *Protogoniomorpha parhassus*, *Hypolimnas bolina*, *Apatura laverna*, and *Charaxes zoolina* (Figure 5). As revealed in Figure 1, the Pieridae, well known to have more or less uniform white or yellow wings, are especially interesting. They generally display the same color in reflected and transmitted light (see also *Colotis regina* and *Colias erate*, Figure 5).

Although only a small dataset is presented here, it highlights the importance of translucence in butterflies. Further evidence sufficiently exists for the hypothesis that dorsal and ventral wing patterns are often quite similar, which can be deduced from published inventories that survey all butterfly families with photographs of both wing sides (e.g., [31,32]).

### 4.2. Different Wing Translucence in Papilionids

The dorsal and ventral wing sides of *Papilio phorcas*, *Graphium sarpedon*, and *G. milon* are marked by a blue-cyan or greenish midband. This color is due to the bile pigments sarpedobilin and phorcabilin [23,28,33]. Midband patterns with very similar colors can be recognized in the dorsal wings of *P. nireus*, *P. bromius*, and *P. epiphorbas*, but here, the color has a mixed pigmentary and structural basis. In these butterflies of the *nireus* group, the upper lamina of the cover scales contains papiliochrome II, the same UV/violet-absorbing pigment that prominently colors *P. xuthus*. It serves as a high-pass color filter in front of the lower lamina, which acts as a thin-film reflector that has a peak reflectance in the blue-wavelength range. The combined pigmentary and structural effects cause the blue-cyan wing color [15]. Brown-black ground scales, which contain a high melanin concentration, together with the heavily-melanized scales of the ventral wing side, serve as backing [15,34]. This backing is essential for absorbing stray light, which otherwise will downgrade a bright coloration. However, it also causes low ventral reflectance and poor wing transmittance, so the dorsal reflection and transmission images of these butterflies strongly differ (Figure 1F and Figure 4G,H). We conclude that closely related species can have very translucent or rather fully opaque wings.

The very translucent hindwings of *P. xuthus* have a row of small patches with blue structural coloration. In transmitted light, those areas are fully opaque, due to melanin backing (Figure 1A). However, the orange patches located distally at both the dorsal and ventral sides of the hindwing are colocalized, and as they also appear in transmitted light, this means that melanin is absent here. The same exact phenomenon can be recognized in the wings of *P. ophidicephalus* (Figure 4F).

The biological function of the distinct midband in the wings encountered in many papilionoid species has been investigated with behavioral assays on *Papilio demolion* Linnaeus, 1758 [35]. The greenish-blue color of the band, created by pigmented scales, was found to support differential blending into the background, whereas the shape of the band hinders the detection of the butterfly through coincident disruptive coloration. Whether the translucence of the band also is also used for enhanced visual signaling needs further attention, but a study on the blue-banded wing pattern of some *Morpho* species, created by structurally colored scales, suggests that it supports escape mimicry, i.e., the color pattern increases the difficulty of capture by predators [36].

### 4.3. Variations in Translucence through Pigments

In the iconic case of *Morpho* butterflies, where multilayered scales on the dorsal wings cause a brilliant-blue structural coloration, a melanin layer is prominently present as an absorbing basis (for instance, in *Morpho didius* [19]). Their ventral wings are colored very differently with a diverse pigmentary pattern (see also *M. menelaus* [37]). Yet, the wings of *M. epistrophus* are rather translucent, and those of *M. cypris* and their close relative *M. rhetenor* have a translucent midband not very different from the midband of papilionid wings [9,38]. The wing membrane of the white, translucent midband of *M. cypris* is unpigmented and transparent, but the wing membrane is melanin-pigmented and opaque exactly where the scales are structurally colored [9].

A large butterfly family with translucent wings is the Pieridae, where the dorsal and ventral wings often have a similar color, due to wing scales colored by pterin pigments that are concentrated in small, strongly light-scattering granules [39]. Visual signaling is enhanced in many male pierids. For instance, in *Hebomoia*, *Colotis*, *Colias*, and *Gonepteryx* species, their dorsal wing tips combine a distinct pigmentation with structural coloration and thus function as special visual signaling devices (Figure 1J, Figure 2F and Figure 5H).

Butterflies with exceptionally translucent wings are the glasswing butterflies Nymphalidae and tribus Ithomiini, which have very reduced scales and in which transparency is even enhanced by nanostructuring of the wing membrane surface [40,41]. As their wing margins and veins are quite opaque, these butterflies presumably have optimized visual signaling under transmitted light.

A most intriguing and obvious question is of course how the patterning of dorsal and ventral wing sides is genetically controlled. In various nymphalids, central roles are attributed to, for instance, *ultrabithorax*, *optix*, *aristaless1*, *aristaless2*, *WntA*, *cortex*, and *wingless* [37,42,43,44], but how these genes determine the outfit of the large variety of butterfly species still requires much research effort.

### 4.4. Biological Implications of Translucence

How the wing translucence is connected to a butterfly’s life history and its habitat and behavior is an intriguing question. Translucent wings will not reduce but enhance the butterfly’s visibility. The enhanced visual signaling will be effective during aerial interactions, especially in species that perform courtship through hill-topping [45,46]. For instance, the mating success of male *Pararge aegeria* was larger when resident in a large sunspot because of its greatly increased ability to detect and intercept passing receptive females [47].

The generally accepted view is that the dorsal wing sides of butterflies are used for display and thermoregulation, while the ventral side patterning favors camouflage, crypsis, or masquerade. The dorsal wing side is indeed commonly more brightly colored than the ventral side. Nevertheless, with a moderate pigment content, resulting in translucent wings, and with similar patterning of the two wing sides, the same pattern is often displayed dorsally in reflected and transmitted light (Figure 1). This will enhance the butterfly’s visual signaling during flight and possibly also when perched with partially open wings. When the wings are closed, the transmittance of the two wings will be much less than that of a single wing. This will especially hold when in rest because the hindwings then usually more or less cover the forewings. As the pattern of the ventral hindwings is often distinctly duller than that of the ventral forewings, the subdued ventral hindwings will be most visible, consistent with a function for camouflage or crypsis [48].

We conclude that there are two conflicting selection mechanisms that are in play: (*i*) the drive to maximize the display, which functions toward sexual and aposematic signaling vs. (*ii*) the drive to be inconspicuous, so as to minimize the chance of being captured, i.e., to enhance survival. Which selection mechanism dominates will depend on the animal’s life history, habitat, and ultimately, its evolution. Whether the wing translucence of a certain butterfly species indeed plays an important role in natural conditions requires targeted, specific behavioral investigations. For instance, we expect that filming and analyzing the behavior of hill-topping male butterflies, both translucent and nontranslucent species, will provide valuable insight. Additionally, behavioral experiments testing the attractivity of suitably painted dummy butterflies should be rewarding. We finally note that the display of numerous other insects with patterned wings, such as flies, dragonflies, and owlflies, will be similar in reflected and transmitted light. Wing translucence thus deserves further entomological study.

## Figures and Tables

**Figure 1 insects-14-00234-f001:**
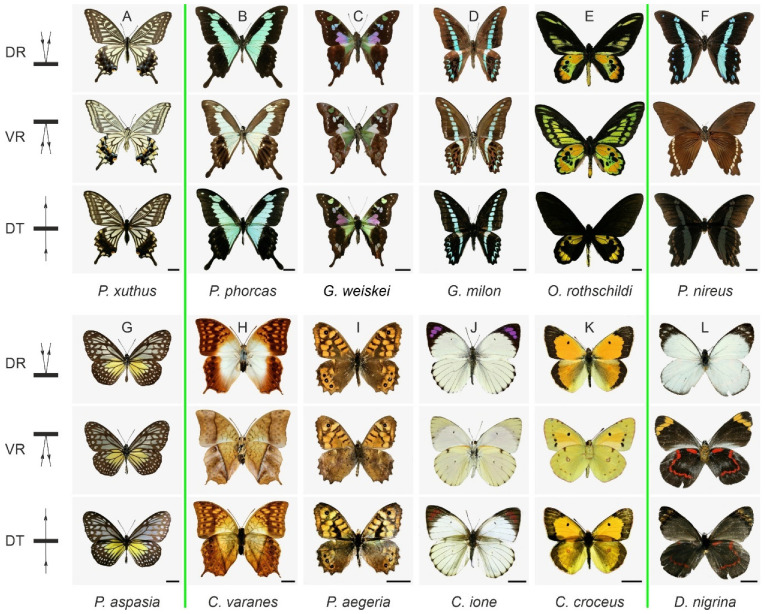
Wing patterning and translucence of butterfly wings: (**A**–**F**) six papilionids: *Papilio xuthus* Linnaeus, 1767; *Papilio phorcas* Cramer, 1775; *Graphium weiskei* Ribbe, 1900; *Graphium milon* Felder and Fleder, 1864; *Ornithoptera rothschildi* Kenrich, 1911; and *Papilio nireus* Linnaeus, 1758; (**G**–**I**) three nymphalids: *Parantica aspasia* Fabricius, 1787; *Charaxes varanes* Cramer, 1777; and *Pararge aegeria* Linnaeus, 1758; (**J**–**L**) three pierids: *Colotis ione* Cramer, 1780; *Colias croceus* Geoffroy, 1785; and *Delias nigrina* Fabricius, 1775. The butterflies in between the green lines are transition cases from very similar dorsal reflection and transmission images (left) to very different images (right). DR—dorsal reflection; VR—ventral reflection; DT—dorsal transmission; scale bars: 1 cm.

**Figure 2 insects-14-00234-f002:**
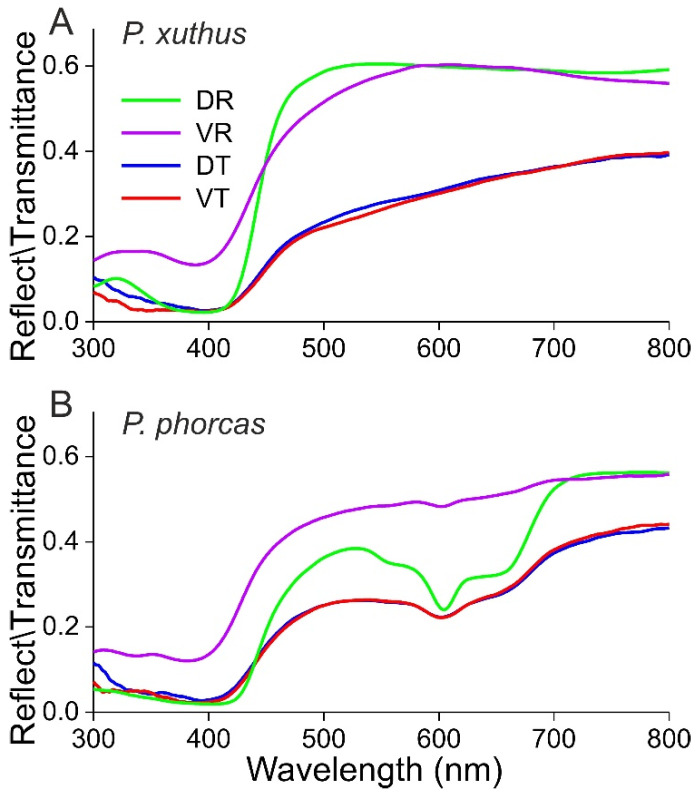
Reflectance and transmittance spectra of butterfly wings: (**A**) *Papilio xuthus* Linnaeus, 1758; (**B**) *Papilio phorcas* Cramer, 1775. DR—dorsal reflection; VR—ventral reflection; DT—dorsal transmission; VT—ventral transmission.

**Figure 3 insects-14-00234-f003:**
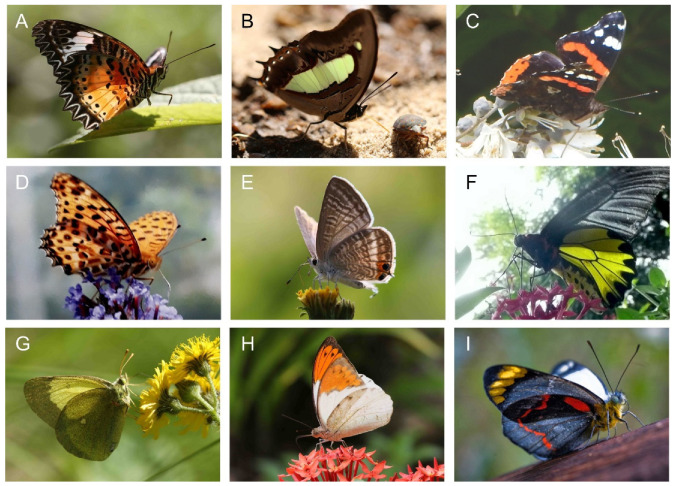
Butterflies with translucent wings illuminated from above: (**A**) *Cethosia penthesilea* Cramer, 1777; (**B**) *Polyura athamas* Drury, 1773; (**C**) *Vanessa atalanta* Linnaeus, 1758; (**D**) *Argyreus hyperbius* Linnaeus, 1763; (**E**) *Lampides boethius* Linnaeus, 1767; (**F**) *Troides aeacus* Felder and Felder, 1860; (**G**) *Colias palaeno orientalis* Staudinger, 1892; (**H**) *Hebomoia glaucippe* Linnaeus, 1758; (**I**) *Delias nigrina* Fabricius, 1775.

**Figure 4 insects-14-00234-f004:**
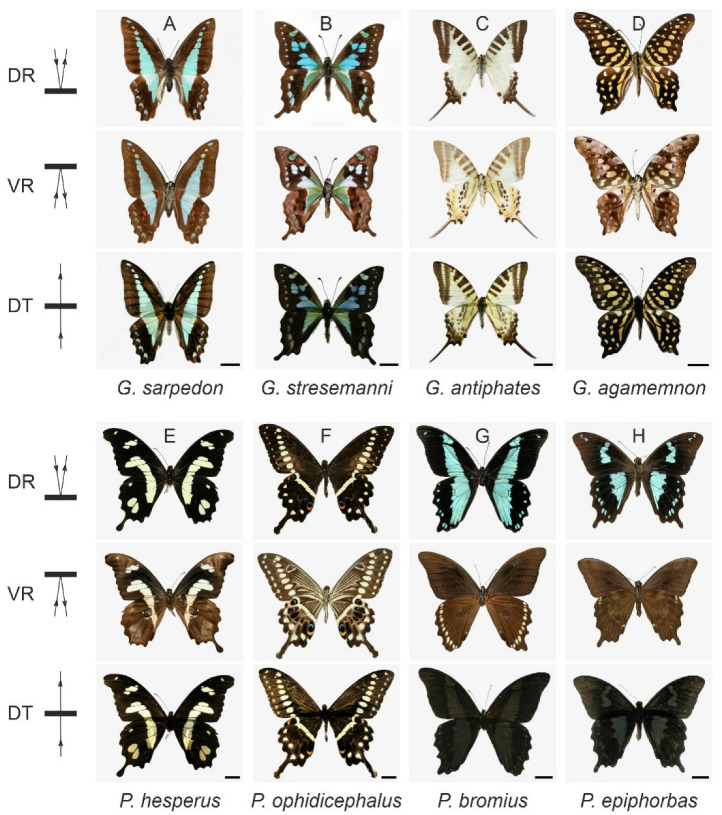
Wing patterning and translucence of papilionid butterfly wings: (**A**) *Graphium sarpedon* Linnaeus, 1758; (**B**) *Graphium stresemanni* Linnaeus, 1758; (**C**) *Graphium antiphates* Cramer, 1775; (**D**) *Graphium agamemnon* Linnaeus, 1758; (**E**) *Papilio hesperus* Westwood, 1843; (**F**) *Papilio ophidicephalus* Oberthür, 1878; (**G**) *Papilio bromius* Doubleday, 1845; (**H**) *Papilio epiphorbas* Boisduval, 1833. DR—dorsal reflection; VR—ventral reflection; DT—dorsal transmission; scale bars: 1 cm.

**Figure 5 insects-14-00234-f005:**
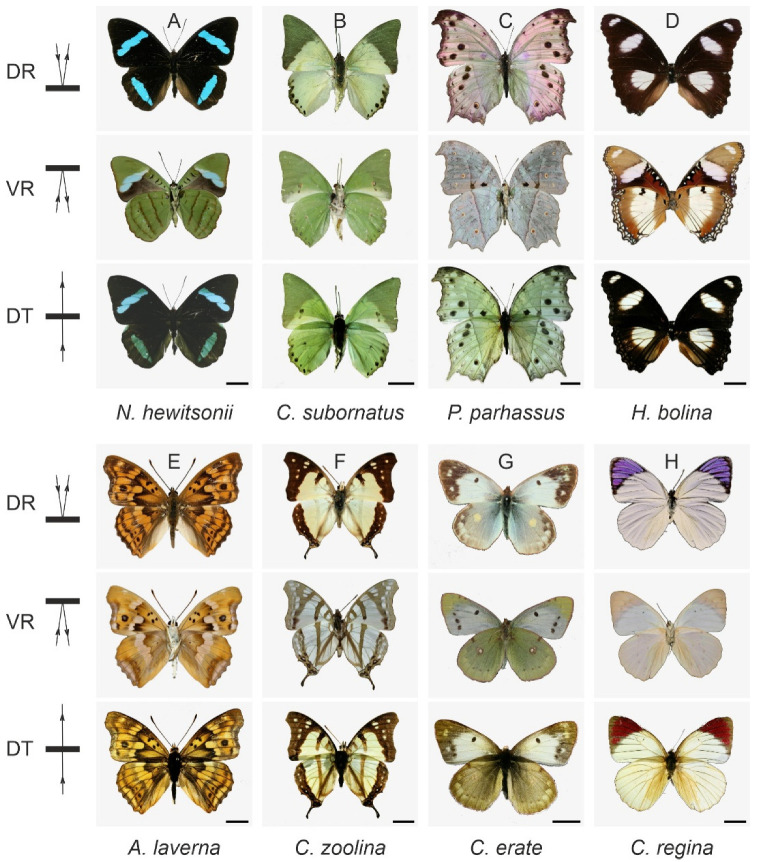
Wing patterning and translucence of six nymphalids and two pierid butterfly wings: (**A**) *Nessaea hewitsonii* Felder and Felder, 1859; (**B**) *Charaxes subornatus* Schultze, 1914; (**C**) *Protogoniomorpha parhassus* Drury, 1782; (**D**) *Hypolimnas bolina* Linnaeus, 1758; (**E**) *Apatura laverna* Leech, 1893; (**F**) *Charaxes zoolina* Westwood, 1850; (**G**) *Colias erate* Esper, 1805; (**H**) *Colotis regina* Trimen, 1863. DR—dorsal reflection; VR—ventral reflection; DT—dorsal transmission; scale bars: 1 cm.

## Data Availability

This article has no additional data.

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
