# Peer review of "Butterfly Wing Translucence Enables Enhanced Visual Signaling"

_insects, 2023, doi:10.3390/insects14030234_

Round 1

Reviewer 1 Report

This is interesting paper, encompassing a truly "new look" at superficially trivial feature of butterfly patterns, namely, that their ventral and dorsal wings occasionally appear identical in transmitted light. You are perfectly right in Introduction that while the so far prevailing views of butterfly patterns oscillate between that of (aposematic) signaling, and crypsis/camouflage, an obvious third option, that both dorsal and ventral side may be used for signaling has rarely, if ever, been investigated. 

As such, I consider the paper a useful addition to the literature and certain inspiration for further and deeper research. So much being said, however, the manuscript needs some small adjustment, to become more readable for people who know much about butterflies, but almost nothing about physics of light and colours. Plus, it needs some minor additions to better conveý with communication style of traditional entomologists, because precisely these people - experts on butterfly life histories, bahaviour, etc. - represent your potential readers. 

line 75: "RGB photos" - embarrassing as it is, I need to think a bit what it is, and memorize grammar school arts lectures. Which is exactly what you do not want to force your readers. So, drop a few explaining words in this case, and elsewhere where you are using non-entomology jargon (e.g., reflectance and transmittance spectra, just below). 

throughout: the first scientific names of each organism should be spelled in full, and a full scientific name has four components: genus, species, authority, and the year of description (as in "Papilio xuthus Linnaeus, 1768"). This pettiness becomes important in treatise on several unrelated species from various regions. 

70-72 (sources of the mounted butterflies). Optimally, you should list localities of origin of the specimens, with a broad precision (country/state) if precise locality is not available. This may assist future work, asi in the case of your specimen of Pararge aegeria. This West Palearctic species has considerable clinal variation in wing pattern, the light dots being whitish in the North and reddish in the South (your specimen). Presenting the origin is important here. 

Captions to Figs. 1, 3, 4: Taxa names should be in italics (as in case of Figure 2 caption).

line 262: the term "glasswing butterflies" should be accompanied by scientific name of the taxon, which is "Nymphalidae, tribus Ithomiini", I believe. Please, check. 

Perhaps most important, and most difficult: In the closing paragraphs, your argument would profit a lot if you sketch how your observations on "maximizing the display, which which functions for sexual and aposematic signaling" could be tested in field. Wild speculations are fully warranted here, as they will certainly enhance the appeal of the paper. 

Author Response

We thank the editor and referees for their positive assessment of our paper.

We added to line 75: "RGB photos" (true-color).

To explain the terms reflectance and transmittance, we added a sentence to section 2.1:

 (Reflectance and transmittance are physical quantities: the fraction of incident light that is reflected and transmitted, respectively; absorbance is the -log10 of the transmittance.)

We have added the entomological details, as requested.

The italics of species names vanished due to conversion of our submitted manuscript into the journal’s format. We hope to have repaired all formatting errors.

Concerning the glasswing butterflies, we added: Nymphalidae, tribus Ithomiini.

The request to ‘sketch how your observations on "maximizing the display, which functions for sexual and aposematic signaling" could be tested in field’ was answered as follows.

After the last sentence except two: Whether the wing translucence of a certain butterfly species indeed plays an important role in natural conditions will require targeted, specific behavioral investigations. For instance, we expect that filming and analyzing the behavior of hilltopping male butterflies, both of translucent and non-translucent species, will provide valuable insight. Also, behavioral experiments testing the attractivity of suitably painted dummy butterflies should be rewarding.

Reviewer 2 Report

This paper makes the case that transmitted light can affect the color output of butterfly wings. The paper is well written, the figures are excellent, and the message simple and well conveyed. I believe this is an interesting piece and I find it suitable for publication in Insects

Author Response

We thank the editor and referees for their positive assessment of our paper.